# Velcrand Functionalized Polyethylene

**DOI:** 10.3390/molecules24050902

**Published:** 2019-03-05

**Authors:** Jonathan Tellers, Jérôme Vachon, Maria Soliman, Enrico Dalcanale, Roberta Pinalli

**Affiliations:** 1Department of Chemistry, Life Sciences and Environmental Sustainability, University of Parma, and INSTM, UdR Parma, Parco Area delle Scienze 17/A, 43124 Parma, Italy; jonathan.tellers@studenti.unipr.it (J.T.); enrico.dalcanale@unipr.it (E.D.); 2SABIC Europe B.V., Urmonderbaan 22, 6160 AH Geleen, The Netherlands; jerome.vachon@sabic.com (J.V.); maria.soliman@sabic.com (M.S.)

**Keywords:** velcrand, polyethylene, quinoxaline cavitand, PE-HEMA

## Abstract

Velcrands are a specific class of cavitands whose complementary surfaces induce self-dimerization. The insertion of a velcrand as physical cross-linking unit into a polymer is reported. To this purpose, the velcrand was functionalized at the lower rim with an isocyanate group. The functional velcrand was reacted with poly (ethylene-co-(2-hydroxethylmethacrylate)) (PE-HEMA), a polymer equipped with free hydroxyl groups suitable for reaction with the isocyanate group. The obtained functionalized polymer was characterized by nuclear magnetic resonance (NMR), differential scanning calorimetry (DSC), and Fourier transform infrared spectroscopy (FTIR), proving the introduction of velcraplexes in the polymer. Films with varying amounts of velcrands were obtained by solution casting and slow evaporation, testifying the processability of the functionalized polymers. The obtained films were used to measure the oxygen barrier properties of the functionalized material.

## 1. Introduction

Cavitands are programmable synthetic receptors capable of hosting shape complementary guests through specific weak interactions, such as hydrogen bonding, π-π stacking, and CH-π and cation-π interactions [1]. Their remarkable and versatile molecular recognition properties have been exploited in many different fields, including catalysis [2,3,4,5], crystal engineering [6], molecular grippers [7], amino acids [8,9] and protein recognition [10,11], responsive nanostructures [12,13], and sensing [14,15].

Among the cavitand family, quinoxaline cavitands occupy a unique position due to their peculiar ability to change their conformation in response to external stimuli, like pH and temperature [16]. In fact, these macrocycles are able to reversibly interconvert between a closed vase conformation and an open extended kite one [17]. The introduction of four methyl groups at the upper rim of the resorcinarene scaffold stabilizes the kite conformer and induces its dimerization to form velcaplexes [18].

Early attempts to turn velcrands into polymeric materials relied on solvophobic interactions (π-π stacking) present between quinoxaline cavitands in the kite form [18], either as heteroditopic or homoditopic monomers. In the first case, the second interaction mode was a metal coordination between pyridines inserted at the lower rim of the kite cavitand and suitable Pd complexes precursors [19,20] or a ureidopyrimidone (UPy) unit which self-dimerizes via quadruple H-bonding [21]. In the second case, two velcrands were covalently linked to form an homoditopic monomer [22]. In all cases the interaction strength was sufficient to generate oligomers, but not polymers. No attempts were made so far to insert velcraplexes into preformed polymers as physical cross-linking units.

Here we report the synthesis and characterization of a velcrand-grafted polyethylene. Polyethylene is one of the most important polymers world-wide, thus any research effort to improve their functionality, like the introduction of physical cross-links, is highly desired. We foresee potential benefits regarding the oxygen barrier properties of functionalized films thanks to the supramolecular cross-linking. Typically, the major contribution to the oxygen barrier properties of polymers is the tortuous pathway that oxygen will have to travel inside the polymer matrix [23]. Here, the considerably large velcrand molecules could provide an obstacle for the oxygen traveling inside the matrix, effectively increasing the barrier properties by additional physical cross-linking, like hydrogen-bonding in ethylene vinyl alcohol (EVOH) but without humidity interference.

## 2. Results and Discussion

For the preparation of velcrand cross-linked polyolefins, polymers bearing moieties suitable for grafting of the velcrands through a reliable, high yield reaction are required. Urethane was chosen as the linking unit because it can be easily obtained by reaction of an isocyanate with a primary alcohol. Poly (ethylene-co-(2-hydroxethylmethacrylate)) (PE-HEMA), a co-polymer of ethylene and 2-hydroxyethylmethacrylate (HEMA), was selected for this purpose. It has free pendant primary hydroxyl groups which can easily react with isocyanate functionalized velcrands. HEMA content is only 0.91 mol%, so to be as close as possible to a true polyethylene.

A self-dimerizing velcrand with a relatively large K_a_ of 8.5·10^4^ M^−1^ (measured in chloroform at room temperature) was employed in this study (Figure 1, Scheme 1) [18,24]. The velcrand is functionalized at the upper rim with four methylene groups, which forces it to assume a permanent kite conformation (velcrand) suitable for dimer formation. The driving forces for dimerization are dipole–dipole, van der Waals, and solvophobic interactions [18]. For polymer insertion, the velcrand was functionalized at the lower rim with an isocyanate group that is suitable for reaction with the free hydroxyl pendant groups of PE-HEMA.

### 2.1. Design and Synthesis of the Velcrand

A resorcinarene functionalized at the lower rim with a hydroxyl group was prepared by reacting 4 equivalents of 2-methylresorcinol with 1 equivalent of 2,3-dihydrofuran and 3 equivalents of butyraldehyde, in methanol as solvent, and concentrated hydrochloric acid. This procedure leads to a mixture of resorcinarenes with zero to two hydroxyl groups at the lower rim. The crude resorcinarene mixture was directly used in the next step, eliminating an additional purification. The crude was reacted overnight at 80 °C with a slight excess of 2,3-dichloroquinoxaline in dimethylformamide (DMF), and in the presence of potassium carbonate as base. The desired velcrand equipped with a single hydroxyl group at the lower rim, hitherto referred to as 4QxCavOH (Scheme 1), was isolated via column chromatography. To obtain the desired isocyanate functional velcrand, 4QxCavOH was reacted with an excess of hexamethylene diisocyanate (HDI) in the presence of dibutyltin dilaurate (DBTDL) acting as a catalyst. The isocyanate functionalized velcrand, namely 4QxCavNCO, was isolated by precipitation in hexane with a yield of 69% (Scheme 1).

### 2.2. Synthesis and Characterization of Velcrand Functional Polymer Networks

PE-HEMA with 0.91 mol% hydroxyl groups was selected as suitable polymer. The functionalized polymers were obtained by reacting PE-HEMA with the isocyanate functional velcrand according to the following procedure. PE-HEMA was dissolved in toluene at 80 °C, then 4QxCavNCO was added (in wt%), followed by the addition of few drops of DBTDL as catalyst. The resulting solution was stirred for 4 h and subsequently precipitated into hexane, obtaining the functionalized polymer as white powder. An overview of the obtained polymers, the introduced amount of velcrand in wt%, and DSC data are given in Table 1. The maximum amount of introduced velcrand, namely 30.6 wt%, corresponds to full functionalization of all the available PE-HEMA hydroxyl groups (0.91 mol%). Introduction of the velcrand proved to affect the thermal properties of the polymer (Table 1 and Figure 2). The bulky motif reduced the T_m_ and the crystalline degree (X_c_) of the pristine polymers, this last one till less than half value for 4QxCav30.6. 

This observation was not unsuspected, as the bulky velcrand side chains cannot be part of the PE crystal lattice and will keep large part of the chain unavailable for crystallization. Small amounts of velcrand do affect the crystallinity to a lesser extent, as is apparent from sample 4QxCav1 (entry 2, Table 1).

The successful synthesis was confirmed via FTIR and high temperature ^1^H NMR. In Figure 3 the FTIR spectra of 4QxCavNCO, pristine and functional PE-HEMA are shown. An absorption band for the OH groups of the HEMA co-monomer is missing in the spectrum of the pristine sample, likely because of the low HEMA content of 0.91%. After reaction, the disappearance of the characteristic NCO stretching band at 2269 cm^−1^ [26] was observed, indicating the reaction of the NCO groups. In addition, new peaks between 1000 cm^−1^ and 1400 cm^−1^ appeared inside the spectrum of the functional polymer labeled 4QxCav30.6 (entry 4, Table 1), which correspond to vibrations of the introduced velcrand. The C=O band resulting from the carbamate linkage is overlapping with the preexisting carbonyl C=O band of the ester of the pristine polymer. 

The attachment of the velcrand to the polymer backbone is confirmed by high temperature ^1^H NMR. The ^1^H NMR of pristine PE-HEMA (top spectrum, blue line), 4QxCav30.6 polymer (middle spectrum, red line), and pristine velcrand 4QxCavNCO (bottom spectrum, green line) are reported in Figure 4. After functionalization, the peaks of pristine PE-HEMA, highlighted by letters **a** and **b** (top spectrum, blue line), combine to a single peak (signal with letter **c** in the middle spectrum, red line). The signal of methylene close to isocyanate marked by letter **g** (bottom spectrum, green line) disappears after functionalization, indicating full functionalization. Due to the overlapping of velcrand and pristine PE-HEMA signals, determination of degree of functionalization was difficult. The ^1^H NMR spectrum of functionalized 4QxCav30.6 (middle spectrum, red line) reveals that the dimer exists in the kite form inside the polymer matrix as the characteristic split of the arylmethyl signals is observed (marked by letter **h** in the bottom spectrum of the pristine velcrand and letter **d** in the middle spectrum of the functionalized polymer) [18]. It is worth noticing that the velcraplex formation is effective even once attached to the PE matrix.

To test the processability, solution casting technique was employed. Firstly, the functionalized polymers were dissolved in tetrachloroethane (TCE) (Sigma Aldrich, St. Louis, MO, USA) at 100 °C, then the obtained solutions were transferred into a polytetrafluoroethylene (PTFE) mold and the solvent was slowly evaporated at 120 °C. Thereby, thin transparent films with a thickness of around 100 μm were obtained (Figure 5). 

The films were subsequently used to determine the oxygen barrier properties of pristine and velcrand functionalized films. The oxygen transmission rate for pristine and velcrand functionalized PE-HEMA as a function of velcrand content in wt% is given in Figure 6A. Clearly, the introduction of the velcrand into the polymers leads to a significant increase in the oxygen transmission rate, therefore decreasing its barrier property towards oxygen. This is most likely caused by the decreased crystallinity of functionalized samples (see Table 1). Despite the small decrease in X_c_ for materials with 1 wt% and 5 wt% of velcrand, a significant decrease in barrier property was observed. This can likely be explained by the increase in free volume caused by the bulky velcrand, which also adversely affects the oxygen barrier properties of polymers [23]. Sample 4QxCav1 presents an outlier with a far greater increased transmission than the sample with higher velcrand content, and additional defects in the sample are likely responsible. 

Additionally, the effect of humidity on oxygen transmission was investigated, measuring oxygen transmission at 0%, 50%, and 100% relative humidity. The results of these measurements for pristine polymer (4Qx-Cav0) and 4QxCav30.6 are given in Figure 6B. No influence of humidity was apparent for either pristine or functionalized materials, showing the resilience of these materials towards water. This was unexpected in the case of PE-HEMA, as hydrogen bonds of HEMA should be disturbed by increased humidity. Most likely, the apolar nature of PE backbone chains shields the hydrogen bonds of HEMA from influences of increased humidity for this polymer with low HEMA content.

## 3. Experimental

Unless otherwise specified, chemicals and solvents were purchased from Sigma-Aldrich and used as received. *N,N*-Dimethylformamide (DMF) was dried over 4Å molecular sieves prior to use. Dichloromethane (DCM) was distilled and stored over 4Å molecular sieves. All other employed solvents were laboratory grade and were used as received. The PE-HEMA polymer used in this study was provided by SABIC (M_n_ = 16,200 g/mol, M_w_ = 63,000 g/mol, Dispersity = 3.9, and 0.91 mol% of HEMA). The starting mono-hydroxyl methyl resorcinarene and the 4QxCavOH were prepared following published procedures [19,27].

### 3.1. Proton Nuclear Magnetic Resonance (^1^H-NMR)

^1^H-NMR-spectra were recorded on a Bruker Avance 400 (400 MHz) spectrometer (Bruker, Billerica, MA, USA) and a Bruker Avance 300 (300 MHz) at room temperature. Chemical shifts are reported in ppm. ^1^H-NMR chemical shifts are given in reference to the residual solvent peak at 7.26 ppm in CD_2_Cl_2_ and at 2.50 ppm in DMSO-d_6_. High temperature nuclear magnetic resonance (HTNMR) spectra were recorded on a Bruker AVANCE III (400 MHz) (Bruker, Billerica, MA, USA) equipped with a cryogenically cooled probe head at 100 °C and 120 °C in TCE-d_2_. ^1^H-NMR chemical shifts are reported in ppm and given in reference to the residual solvent peak of TCE-d_2_ at 6.00 ppm.

### 3.2. Fourier Transformed Infrared Spectroscopy (FTIR)

FTIR-spectra were recorded on a Perkin Elmer Spectrum One equipped with a Golden Gate accessory (diamond ATR) (Perkin Elmer, Waltham, MA, USA).

### 3.3. Differential Scanning Calorimetry (DSC)

The DSC measurements were performed on a TA Instruments Q20 (TA Instrument, New Castle, DE, USA) equipped with a RCS 90 cooling system. About 3–5 mg of polymeric sample was weighed inside an aluminum pan and subjected to DSC measurements under nitrogen atmosphere. Unless otherwise noted, polymers were screened twice from −40 °C to 200 °C at a constant heating/cooling rate of 10 K·min^−1^. The T_m_, and the melting enthalpy (H_m_) were determined from the second heating.

### 3.4. Oxygen and Water Transmission

Oxygen transmission rates were determined according to ISO 15,105 2 at 23 °C, and at 0% r.h., 50% r.h., and at 100% r.h. The thickness of films, which was around 100 μm, was measured at varying locations and the average was used to report thickness corrected oxygen transmission rates.

### 3.5. Synthesis of 4QxCavNCO

The reaction was performed under dry condition. HDI (0.5 mL, 3.11 mmol, 8 eq) was inserted into a Shlenk flask and degassed. Subsequently, dichloromethane (25 mL) was added and the mixture cooled to 0 °C. Then, 4QxCavOH (0.5 g, 0.41 mmol, 1 eq), dissolved in DCM (50 mL), was added via syringe, followed by the addition of a drop of DBTDL as catalyst. The reaction was stirred overnight at room temperature. The solute was precipitated in cold (0 °C) diethylether, filtered, dried, redissolved in DCM and then precipitated into pentane. The obtained precipitate was filtered and dried, obtaining the desired product as a white powder (0.39 g, 69% yield). ^1^H-NMR (300 MHz, CD_2_Cl_2_) δ: 7.78–7.71 (m, 4H), 7.70–7.63 (m, 4H), 7.52–7.44 (m, 4H), 7.15 (s, 4H), 6.90 (s, 2H), 6.19 (s, 2H), 4.86 (s, 1H), 4.1–3.74 (m, 2H), 3.71–3.5 (m, 4H), 3.25 (s, 2H), 3.07 (s, 6H), 3.02–2.84 (m, 2H), 2.20 (s, 6), 2.09–1.68 (m, 8H) 1.66–0.97 (m, 16H), 0.73 (t, J = 7.2 Hz, 9H). MS (MALDI) m/z: [M + H]^+^ calcd for C_84_H_77_N_10_O_11_, 1401.58; found, 1401.41.

### 3.6. Synthesis of Velcrand Functionalized Polyethylene

Polymer functionalization with velcrand was carried out according to the following general procedure, exemplified for 4QxCav30.6: The reaction was performed under dry conditions. 

PE-HEMA (0.5 g) with 0.91% of HEMA groups, was dissolved in toluene (50 mL) at 80 °C. Then, the 4QxCavNCO (0.28 g, 1 eq with respect to OH groups) was added as a solid. After dissolution of the velcrand, two drops of DBTDL were added to function as a catalyst. The mixture was stirred at 80 °C for 4 h. The reaction was quenched in acetone, resulting in precipitation of a white powder. Filtration and drying of the precipitate gave the product as a white powder.

## 4. Conclusions

A self-dimerizing velcrand in kite conformation functionalized at the lower rim with an isocyanate group was synthesized. This velcrand, namely 4QxCavNCO, was grafted onto PE-HEMA, a polymer with free hydroxyl groups, via urethane bond formation. The successful synthesis of these functionalized polyolefins was confirmed via FTIR and ^1^H-NMR spectroscopy. Transparent films of functional polymers were obtained by solution casting and slow evaporation of the solvent. Thereby obtained films were tested for their oxygen transmission, which was found to sequentially increase with increasing velcrand concentration. This lackluster performance can be a result of the severely reduced degree of crystallinity (*X*_c_) and increased free volume due to the introduction of the bulky velcrand, suggesting that the expected increased interfacial cohesion between chains in the amorphous phase, which is associated with increased barrier properties, did not occur. 

Most significantly, 4QxCavNCO velcrand provides a tool to introduce reversible supramolecular cross-linking into polymers [19].

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
