# Peer review of "Velcrand Functionalized Polyethylene"

_molecules, 2019, doi:10.3390/molecules24050902_

Round 1
Reviewer 1 Report
After carefully read this manuscript, I would like to make some comments about its content:
First, the authors should correct the scheme 1, for example there are some mistakes as "HCL" or Nitrogen missing in the structure of the 2,3-dichloroquinoxaline. Maybe it should be better to modify the figure 3 for the understanding of the readers (when I printed the document in black and white it appeared difficult to discriminate each peaks).
Maybe the authors could measure XRD of their materials to prove the influence of the velcrand on the cristallinity of their products.
Is it possible to add the respective thermograms in this manuscript ?
Author Response
We thank the Referee for the suggestions that helped us to improve the manuscript. We followed the comments and suggestions to revise our manuscript. The paper was adapted accordingly, and punctual responses to the comments are listed below.
After carefully read this manuscript, I would like to make some comments about its content:
1. First, the authors should correct the scheme 1, for example there are some mistakes as "HCL" or Nitrogen missing in the structure of the 2,3-dichloroquinoxaline. Maybe it should be better to modify the figure 3 for the understanding of the readers (when I printed the document in black and white it appeared difficult to discriminate each peaks).
We thank the reviewer for having noticed the mistake, the structure of 2,3-dicholorquinoxaline in Scheme 1 was corrected accordingly.
Figure 3 (in the revised manuscript Figure 4) and the relative caption have been updated accordingly.
2. Maybe the authors could measure XRD of their materials to prove the influence of the velcrand on the cristallinity of their products.
We calculated the crystallinity degree (Xc, Table 1) of all the samples to prove the influence of the velcrand on the crystallinity of the product.
3. Is it possible to add the respective thermograms in this manuscript?
Figure 2 reporting the DSC spectra of the samples was added.

Reviewer 2 Report
Recommendation: Publish after major revisions.
Comments: The authors developed a functionalization method by introducing velcrands into polymer PE-HEMA. This method is based on NCO group from velcrand derivatives reacted with OH group in PE-HEMA. This synthesis is straightforward, efficient and possibly applicable to other systems. However, there are a few errors/concerns that should be addressed before publication. Hope the authors reflect the following comments and improve the work.
1. In Page 1 line 16, the Abstract contains duplicating words, i.e., “FTIR FTIR”. Please kindly correct it.
2. Another concern is the Scheme 1 in Page 3 line 78. In the second step, the author used 2,3-dichloroquinoxaline as starting material; however, the drawing of 2,3-dichloroquinoxaline in Scheme 1 is incorrect.
3. In Page 4 Figure 2, the pristine PE-HEMA doesn’t present any observable peak for –OH (3100~3300 cm-1). According to the authors, the PE-HEMA should have 0.91 mol% hydroxyl groups. Can the author provide any reasons?
4. In Page 7 line 188, the author concludes that 4QxCavNCO velcrand provides a tool to 188 introduce reversible supramolecular cross-linking into polymers. However, there is no experiment on “reversible supramolecular cross-linking”. The author should provide experimental evidence to support this claim.
Author Response
We thank the Referee for the suggestions that helped us to improve the manuscript. We followed the comments and suggestions to revise our manuscript. The paper was adapted accordingly, and punctual responses to the comments are listed below.
Reviewer 2:
Comments: The authors developed a functionalization method by introducing velcrands into polymer PE-HEMA. This method is based on NCO group from velcrand derivatives reacted with OH group in PE-HEMA. This synthesis is straightforward, efficient and possibly applicable to other systems. However, there are a few errors/concerns that should be addressed before publication. Hope the authors reflect the following comments and improve the work.
1. In Page 1 line 16, the Abstract contains duplicating words, i.e., “FTIR FTIR”. Please kindly correct it.
We thank the reviewer for having noticed the typo. We have corrected the abstract.
2. Another concern is the Scheme 1 in Page 3 line 78. In the second step, the author used 2,3-dichloroquinoxaline as starting material; however, the drawing of 2,3-dichloroquinoxaline in Scheme 1 is incorrect.
Scheme 1: the structure of 2,3-dicholoroquinoxaline was corrected accordingly.
3. In Page 4 Figure 2, the pristine PE-HEMA doesn’t present any observable peak for –OH (3100~3300 cm-1). According to the authors, the PE-HEMA should have 0.91 mol% hydroxyl groups. Can the author provide any reasons?
We added a comment for this at page 4, lines 111-113.
4. In Page 7 line 188, the author concludes that 4QxCavNCO velcrand provides a tool to introduce reversible supramolecular cross-linking into polymers. However, there is no experiment on “reversible supramolecular cross-linking”. The author should provide experimental evidence to support this claim.
We added reference 19 which supports this claim.

Reviewer 3 Report
Current article reports the synthesis of velcrand functionalized polyethylene. Velcrands functionalized with isocynate group at the lower rim was reacted with PE-HEMA. NMR, DSC, and FTIR were used for analysis of the product. The topic of the study seems attractive. However, I feel there are several points that should be addressed before this article can be accepted for publications
1. The introduction is very short and does not really explain for to the reader the real worth of the topic and what is the state of the art and what is targeted to achieve in this study
2. Result and discussion section contain information that should be in experimental section e.g., lines 65-70, 83-88 and so on
3. Authors gave their explanations referring as colors that would be difficult to follow in printed black and white copy, so must explain in a way that a reader with a black an white copy could follow
4. Figure 3, the protons should be referred as a, b, c etc rather than orange, blue dots
5. Effect of the functionalization on the application properties is not fully elaborated, I would rather recommend to compare for unfunctionalized and functionalized materials for some other properties too
6. The specifications of PE used in study are not given (molar mass etc)
Author Response
We thank the Referee for the suggestions that helped us to improve the manuscript. We followed the comments and suggestions to revise our manuscript. The paper was adapted accordingly, and punctual responses to the comments are listed below.
Reviewer 3:
Current article reports the synthesis of velcrand functionalized polyethylene. Velcrands functionalized with isocynate group at the lower rim was reacted with PE-HEMA. NMR, DSC, and FTIR were used for analysis of the product. The topic of the study seems attractive. However, I feel there are several points that should be addressed before this article can be accepted for publications
1. The introduction is very short and does not really explain for to the reader the real worth of the topic and what is the state of the art and what is targeted to achieve in this study
We added a sentence explaining the target of the work (page 1-2, lines 45-51).
2. Result and discussion section contain information that should be in experimental section e.g., lines 65-70, 83-88 and so on
We prefer to describe the methodology adopted for the synthesis of the materials also in the Result and Discussion section. In fact, the synthesis of some intermediates was not described in the Experimental section since already known in the literature.
3. Authors gave their explanations referring as colors that would be difficult to follow in printed black and white copy, so must explain in a way that a reader with a black and white copy could follow
4. Figure 3, the protons should be referred as a, b, c etc rather than orange, blue dots
Figure 3 (in the revised manuscript Figure 4) and the relative caption have been updated accordingly.
5. Effect of the functionalization on the application properties is not fully elaborated, I would rather recommend to compare for unfunctionalized and functionalized materials for some other properties too
We added the oxygen barrier transmission measurements at pages 5-6 and Figure 6.
6. The specifications of PE used in study are not given (molar mass etc)
We reported the specifications of PE-HEMA in the Experimental section, page 6 lines 176-177.

Round 2
Reviewer 2 Report
From the reading of the revised manuscript, the authors have demonstrated convincingly the scientific value of the research. The revised manuscript has taken care of the issues raised by the reviewers.
I recommend publication of this manuscript in this Journal.
Reviewer 3 Report
Accept